# Supervision Accelerates Pre-training in Contrastive Semi-Supervised Learning of Visual Representations

## Abstract

We investigate a strategy for improving the efficiency of contrastive learning of visual representations by leveraging a small amount of supervised information during pre-training. We propose a semi-supervised loss, SuNCEt , based on noise-contrastive estimation and neighbourhood component analysis, that aims to distinguish examples of different classes in addition to the self-supervised instance-wise pretext tasks. On ImageNet, we find that SuNCEt can be used to match the semi-supervised learning accuracy of previous contrastive approaches while using less than half the amount of pre-training and compute. Our main insight is that leveraging even a small amount of labeled data during pre-training, and not only during fine-tuning, provides an important signal that can significantly accelerate contrastive learning of visual representations.

## 1 Introduction

Learning visual representations that are semantically meaningful with limited semantic annotations is a longstanding challenge with the potential to drastically improve the data-efficiency of learning agents. Semi-supervised learning algorithms based on contrastive instance-wise pretext tasks learn representations with limited label information and have shown great promise (Hadsell et al., 2006; Wu et al., 2018b; Bachman et al., 2019; Misra & van der Maaten, 2020; Chen et al., 2020a). Unfortunately, despite achieving state-of-the-art performance, these semi-supervised contrastive approaches typically require at least an order of magnitude more compute than standard supervised training with a cross-entropy loss (albeit without requiring access to the same amount of labeled data). Burdensome computational requirements not only make training laborious and particularly time- and energy-consuming; they also exacerbate other issues, making it more difficult to scale to more complex models and problems, and potentially inducing significant carbon footprints depending on the infrastructure used for training (Henderson et al., 2020).

In this work, we investigate a strategy for improving the computational efficiency of contrastive learning of visual representations by leveraging a small amount of supervised information during pre-training. We propose a semi-supervised loss, SuNCEt , based on noise-contrastive estimation (Gutmann & Hyvärinen, 2010) and neighbourhood component analysis (Goldberger et al., 2005), that aims at distinguishing examples of different classes in addition to the self-supervised instance-wise pretext tasks. We conduct a case-study with respect to the approach of Chen et al. (2020a) on the ImageNet (Russakovsky et al., 2015) and CIFAR10 (Krizhevsky & Hinton, 2009) benchmarks. We find that using any available labels during pre-training (either in the form of a cross-entropy loss or SuNCEt ) can be used to reduce the amount of pre-training required. Our most notable results on ImageNet are obtained with SuNCEt , where we can match the semi-supervised learning accuracy of previous contrastive approaches while using less than half the amount of pre-training and compute, and require no hyper-parameter tuning. By combining SuNCEt with the contrastive SwAV method of Caron et al. (2020), we also achieve state-of-the-art top-5 accuracy on ImageNet with 10% labels, while cutting the pre-training epochs in half.

## 2 BACKGROUND

The goal of contrastive learning is to learn representations by comparison. Recently, this class of approaches has fueled rapid progress in unsupervised representation learning of images through self-supervision (Chopra et al., 2005; Hadsell et al., 2006; Bachman et al., 2019; Oord et al., 2018; Hénaff et al., 2019; Tian et al., 2019; Misra & van der Maaten, 2020; He et al., 2019; Arora et al., 2019; Chen et al., 2020a; Caron et al., 2020; Grill et al., 2020; Chen et al., 2020b). In that context, contrastive approaches usually learn by maximizing the agreement between representations of different views of the same image, either directly, via instance discrimination, or indirectly through, cluster prototypes. Instance-wise approaches perform pairwise comparison of input data to push representations of similar inputs close to one another while pushing apart representations of dissimilar inputs, akin to a form of distance-metric learning.

Self-supervised contrastive approaches typically rely on a data-augmentation module, an encoder network, and a contrastive loss. The data augmentation module stochastically maps an image $\boldsymbol{x}_i \in \mathbb{R}^{3 \times H \times W}$ to a different view. Denote by $\hat{\boldsymbol{x}}_{i,1}, \hat{\boldsymbol{x}}_{i,2}$ two possible views of an image $\boldsymbol{x}_i$, and denote by $f_\theta$ the parameterized encoder, which maps an input image $\hat{\boldsymbol{x}}_{i,1}$ to a representation vector $\boldsymbol{z}_{i,1} = f_\theta(\hat{\boldsymbol{x}}_{i,1}) \in \mathbb{R}^d$. The encoder $f_\theta$ is usually parameterized as a deep neural network with learnable parameters $\theta$. Given a representation $\boldsymbol{z}_{i,1}$, referred to as an anchor embedding, and the representation of an alternative view of the same input $\boldsymbol{z}_{i,2}$, referred to as a positive sample, the goal is to optimize the encoder $f_\theta$ to output representations that enable one to easily discriminate between the positive sample and noise using multinomial logistic regression. This learning by picking out the positive sample from a pool of negatives is in the spirit of noise-contrastive estimation (Gutmann & Hyvärinen, 2010). The noise samples in this context are often taken to be the representations of other images. For example, suppose we have a set of images $(\boldsymbol{x}_i)_{i \in [n]}$ and apply the stochastic data-augmentation to construct a new set with two views of each image, $(\hat{\boldsymbol{x}}_{i,1}, \hat{\boldsymbol{x}}_{i,2})_{i \in [n]}$. Denote by $\mathcal{Z} = (\boldsymbol{z}_{i,1}, \boldsymbol{z}_{i,2})_{i \in [n]}$ the set of representations corresponding to these augmented images. Then the noise samples with respect to the anchor embedding $\boldsymbol{z}_{i,1} \in \mathcal{Z}$ are given by $\mathcal{Z} \backslash \{\boldsymbol{z}_{i,1}, \boldsymbol{z}_{i,2}\}$. In this work, we minimize the normalized temperature-scaled cross entropy loss (Chen et al., 2020a) for instance-wise discrimination

$$\ell_{\text{inst}}(\boldsymbol{z}_{i,1}) = - \log \frac{\exp(\text{sim}(\boldsymbol{z}_{i,1}, \boldsymbol{z}_{i,2})/\tau)}{\sum_{\boldsymbol{z} \in \mathcal{Z} \backslash \{\boldsymbol{z}_{i,1}\}} \exp(\text{sim}(\boldsymbol{z}_{i,1}, \boldsymbol{z})/\tau)}, \tag{1}$$

where $\text{sim}(a, b) = \frac{a^T b}{\|a\| \|b\|}$ denotes the cosine similarity and $\tau > 0$ is a temperature parameter.

In typical semi-supervised contrastive learning setups, the encoder $f_\theta$ is learned in a fully unsupervised pre-training phase. The goal of this pre-training is to learn a representation invariant to common data augmentations (cf. Hadsell et al. (2006); Misra & van der Maaten (2020)) such as random crop/flip, resizing, color distortions, and Gaussian blur. After pre-training on unlabeled data, labeled training instances are leveraged to fine-tune $f_\theta$, e.g., using the canonical cross-entropy loss.

## 3 METHODOLOGY

Our goal is to investigate a strategy for improving the computational efficiency of contrastive learning of visual representations by leveraging the available supervised information during pre-training. Here we explore a contrastive approach for utilizing available labels, but we also include additional numerical evaluations with a cross-entropy loss and a parametric classifier in Section 4.

**Contrastive approach.** Consider a set $\mathcal{S}$ of labeled samples operated upon by the stochastic data-augmentation module. The associated set of parameterized embeddings are given by $\mathcal{Z}_{\mathcal{S}}(\theta) = (f_\theta(\hat{\boldsymbol{x}}))_{\hat{\boldsymbol{x}} \in \mathcal{S}}$. Let $\hat{\boldsymbol{x}} \in \mathcal{S}$ denote an anchor image view with representation $\boldsymbol{z} = f_\theta(\hat{\boldsymbol{x}})$ and class label $y$. By slight overload of notation, denote by $\mathcal{Z}_y(\theta)$ the set of embeddings for images in $\mathcal{S}$ with class label $y$ (same class as the anchor $\boldsymbol{z}$). We define the _Supervised Noise Contrastive Estimation_ (SuNCEt ) loss as

$$\ell(\boldsymbol{z}) = - \log \frac{\sum_{\boldsymbol{z}_j \in \mathcal{Z}_y(\theta)} \exp(\text{sim}(\boldsymbol{z}, \boldsymbol{z}_j)/\tau)}{\sum_{\boldsymbol{z}_k \in \mathcal{Z}_{\mathcal{S}}(\theta) \backslash \{\boldsymbol{z}\}} \exp(\text{sim}(\boldsymbol{z}, \boldsymbol{z}_k)/\tau)}, \tag{2}$$

which is then averaged over all anchors $\frac{1}{|\mathcal{S}|} \sum_{\boldsymbol{z} \in \mathcal{Z}_{\mathcal{S}}(\theta)} \ell(\boldsymbol{z})$.

In each iteration of training we sample a few unlabeled images to compute the self-supervised instance-discrimination loss equation 1, and sample a few labeled images to construct the set $\mathcal{S}$ and compute the SuNCEt loss equation 2. We sum these two losses together and backpropagate through the encoder network. By convention, when "sampling unlabeled images," we actually sample images from the entire training set (labeled and unlabeled). This simple procedure bears some similarity to unsupervised data augmentation (Xie et al., 2019), where a supervised cross-entropy loss and a parametric consistency loss are calculated at each iteration.

**Motivation.** We motivate the form of the SuNCEt loss by leveraging the relationship between contrastive representation learning and distance-metric learning. Specifically, the SuNCEt loss can be seen as a form of neighborhood component analysis (Goldberger et al., 2005) with an alternative similarity metric. Consider a classifier that predicts an image's class based on the similarity of the image's embedding $z$ to those of other labeled images $z_j$ using a temperature-scaled cosine similarity metric $d(z, z_j) = z^T z_j / (\|z\| \|z_j\| \tau)$. Specifically, let the classifier randomly choose one point as its neighbour, with distribution as described below, and adopt the neighbour's class. Given the query embedding $z$, denote the probability that the classifier selects point $z_j \in \mathcal{Z}_{\mathcal{S}}(\theta) \backslash \{z\}$ as its neighbour by

$$p(z_j | z) = \frac{\exp(d(z, z_j))}{\sum_{z_k \in \mathcal{Z}_{\mathcal{S}(\theta)} \backslash \{z\}} \exp(d(z, z_k))}.$$

Under mutual exclusivity (since the classifier only chooses one neighbour) and a uniform prior, the probability that the classifier predicts the class label $\hat{y}$ equal to some class $c$, given a query image $x$ with embedding $z$, is

$$p(\hat{y} = c | z) = \sum_{z_j \in \mathcal{Z}_c(\theta)} p(z_j | z) = \frac{\sum_{z_j \in \mathcal{Z}_c(\theta)} \exp(d(z, z_j))}{\sum_{z_k \in \mathcal{Z}_{\mathcal{S}}(\theta) \backslash \{z\}} \exp(d(z, z_k))}, \tag{3}$$

where $\mathcal{Z}_c(\theta) \subset \mathcal{Z}_{\mathcal{S}}(\theta)$ is the set of embeddings of labeled images from class $c$. Minimizing the KL divergence between $p(\hat{y}|z)$ and the true class distribution (one-hot vector on the true class $y$), one arrives at the SuNCEt loss in equation 2. Assuming independence between labeled samples, the aggregate loss with respect to all labeled samples $\mathcal{S}$ decomposes into the simple sum $\sum_{z \in \mathcal{Z}_{\mathcal{S}}(\theta)} \ell(z)$. Numerical experiments in Appendix G show that using SuNCEt during pre-training optimizes this aforementioned non-parametric stochastic nearest-neighbours classifier and significantly out-performs inference with the more common K-Nearest Neighbours strategy.

**Practical considerations.** Rather than directly using the outputs of encoder $f_\theta$ to contrast samples, we feed the representations into a small multi-layer perceptron (MLP), $h_{\theta_{\text{proj}}}$, to project the representations into a lower dimensional subspace before evaluating the contrastive loss, following Chen et al. (2020a). That is, instead of using $z = f_\theta(\hat{x})$ directly in equation 1 and equation 2, we use $h_{\theta_{\text{proj}}}(z) = h_{\theta_{\text{proj}}}(f_\theta(\hat{x}))$. The projection network $h_{\theta_{\text{proj}}}$ is only used for optimizing the contrastive loss, and is discarded at the fine-tuning phase. In general, adding SuNCEt to a pre-training script only takes a few lines of code. See Listing 2 in Appendix A for the pseudo-code used to compute SuNCEt loss on a mini-batch of labeled images.

## 4 EXPERIMENTS

In this section, we investigate the computational effects of SuNCEt when combined with the Sim-CLR self-supervised instance-wise pretext task defined in Section 2.[1] We report results on the ImageNet (Russakovsky et al., 2015) and CIFAR10 (Krizhevsky & Hinton, 2009) benchmarks for comparison with related work. We also examine the combination of SuNCEt with the contrastive SwAV method of Caron et al. (2020) in Section 5, and achieve state-of-the-art top-5 accuracy on ImageNet with 10% labels, while cutting the pre-training epochs in half. All methods are trained using the LARS optimizer (You et al., 2017) along with a cosine-annealing learning-rate schedule (Loshchilov & Hutter, 2016). The standard procedure when evaluating semi-supervised learning methods on these data sets is to assume that some percentage of the data is labeled, and treat the rest of the data as unlabeled. On ImageNet we directly use the same 1% and 10% data splits used

---

[1]The SuNCEt loss can certainly be combined with other instance-wise pretext tasks as well.

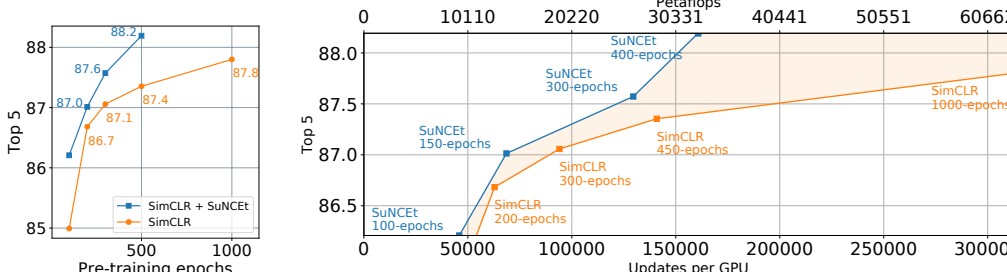

Figure 1: Top-5 validation accuracy of a ResNet50 pre-trained on ImageNet with access to 10% of the labels. Orange markers depict SimCLR self-supervised pre-training followed by fine-tuning. Blue markers depict the combination of SimCLR + SuNCEt . Using SuNCEt to leverage available labels during pre-training (not only fine-tuning), **(i)** accelerates convergence and produces better models (left sub-figure); and **(ii)** can match the semi-supervised learning accuracy of SimCLR whith much less pre-training (right sub-figure). Orange shading in the right sub-figure depicts compute saved. We train all methods using 64 V100 GPUs. One SimCLR epoch corresponds to 312 updates per GPU.

Table 1: Validation accuracy of a ResNet50 pre-trained on ImageNet with access to 10% of the labels (left table) and 1% of labels (right table). All SimCLR implementations are pre-trained for a certain number of epochs and then fine-tuned on the available labels. Using SuNCEt to leverage available labels during pre-training (not only fine-tuning) accelerates training, and produces better models with much less compute.

| 10% Labeled | | Accuracy (%) | | 1% Labeled | | Accuracy (%) | |
|---|---|---|---|---|---|---|---|
| **Method** | **Epochs** | **Top 1** | **Top 5** | **Method** | **Epochs** | **Top 1** | **Top 5** |
| *Chen et al. (2020a) implementation* | | | | *Chen et al. (2020a) implementation* | | | |
| SimCLR | 1000 | 65.6 | 87.8 | SimCLR | 1000 | 48.3 | 75.5 |
| *Our re-implementation* | | | | *Our re-implementation* | | | |
| SimCLR | 1000 | 66.1 | 87.8 | SimCLR | 1000 | **50.8** | **77.7** |
| SimCLR | 500 | 65.4 | 87.3 | SimCLR | 500 | 49.4 | 76.9 |
| + SuNCEt (ours) | 500 | **66.7** | **88.2** | + SuNCEt (ours) | 500 | 49.8 | 77.5 |

by Chen et al. (2020a). On CIFAR10, we create the labeled data sets by independently selecting each point to be in the set of labeled training points with some probability $p$; we run experiments for each $p$ in $\{0.01, 0.05, 0.1, 0.2, 0.5, 1.0\}$.

**Architecture & data.** The encoder network in our experiments is a ResNet-50. On CIFAR10 we modify the trunk of the encoder following Chen et al. (2020a). While this network may not be optimal for CIFAR10 images, it enables fair comparison with previous work. For the projection network $h_{\theta_{\text{proj}}}$ we use an MLP with a single hidden-layer; the hidden layer has 2048 units and the output of the projection network is a 128-dimensional real vector. The stochastic data augmentation module employs random cropping, random horizontal flips, and color jitter. On ImageNet, we also make use of Gaussian blur.

**Fine-tuning.** Upon completion of pre-training, all methods are fine-tuned on the available set of labeled data using SGD with Nesterov momentum (Sutskever et al., 2013). We adopt the same fine-tuning procedure as Chen et al. (2020a). Notably, when fine-tuning, we do not employ weight-decay and only make use of basic data augmentations (random cropping and random horizontal flipping). Additional details on the fine-tuning procedure are provided in Appendix B.

### 4.1 IMAGENET

**Experimental setup.** Our default setup on ImageNet makes use of distributed training; we train each run on 64 V100 GPUs and 640 CPU cores. We aggregate gradients using the standard all-reduce primitive and contrast representations across workers using an efficient all-gather primitive. We also synchronize batch-norm statistics across all workers in each iteration to prevent the models

from leaking local information to improve the loss without improving representations (cf. Chen et al. (2020a)). We linearly warm-up the learning-rate from $0.6$ to $4.8$ during the first 10 epochs of training and use a cosine-annealing schedule thereafter. We use a momentum value of $0.9$, weight decay $10^{-6}$, and temperature $0.1$. These hyper-parameters are tuned for SimCLR (Chen et al., 2020a), but we also apply them to the SimCLR + SuNCEt combination.

We use a batch-size of 4096 (8192 contrastive samples) for SimCLR; each worker processes 128 contrastive samples per iteration. When implementing the SimCLR + SuNCEt combination, we aim to keep the cost per-iteration roughly the same as the baseline, so we use a smaller unsupervised batch-size. Specifically, each worker processes 88 unlabeled samples per iteration, and 40 labeled samples (sub-sampling 20 classes in each iteration and sampling 2 images from each of the sub-sampled classes). With $10\%$ of the images labeled, we turn off the SuNCEt loss after epoch 250; with $1\%$ of the images labeled, we turn off the SuNCEt loss after epoch 30. We explore the effect of the switch-off epoch and the supervised batch-size (the fraction of labeled data in the sampled mini-batch) in Appendix D, and find the ImageNet results to be relatively robust to these parameters.

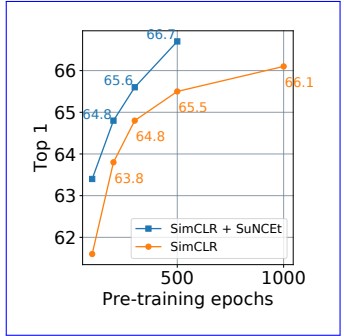

Figure 2: Top-1 accuracy of a ResNet50 pre-trained on ImageNet with access to $10\%$ of the labels. Orange markers depict SimCLR self-supervised pre-training followed by fine-tuning. Blue markers depict the combination of SimCLR + SuNCEt . Using SuNCEt to leverage available labels during pre-training (not only fine-tuning) accelerates convergence and produces better models.

**SuNCEt .**  Figure 1 shows the top-5 accuracy as a function of the amount pre-training when $10\%$ of the data is labeled. Orange markers denote SimCLR self-supervised pre-training followed by fine-tuning. Blue markers denote the SimCLR + SuNCEt combination followed by fine-tuning. Using SuNCEt to leverage available labels during pre-training accelerates convergence and produces better models with much less compute. The orange shaded region in the right sub-figure explicitly shows the amount of compute saved by using SuNCEt during pre-training. To put these results in the context of our 64 GPU setup, one epoch of SimCLR corresponds to 312 updates per GPU. SimCLR+SuNCEt matches the best SimCLR top-5 accuracy while using only $44\%$ of the compute, and matches the best SimCLR top-1 accuracy while using only $45\%$ of the compute. It may be possible to push these savings further by optimizing the hyper-parameters for SuNCEt .

Similarly, Figure 2 shows the top-1 accuracy as a function of the amount pre-training when $10\%$ of the data is labeled. Orange markers denote SimCLR self-supervised pre-training followed by fine-tuning. Blue markers denote the SimCLR + SuNCEt combination followed by fine-tuning. It is interesting to note that the improvements in accuracy are significant under the same training epochs; e.g. in the $10\%$ label setting, with 100 epochs of training we observe a $+1.6\%$ improvement (from $61.8\%$ to $63.4\%$) in top-1 ImageNet accuracy (run-to-run variation is on the order of $0.1$-$0.2\%$); similarly with 500 epochs of training we observe a $+1.3\%$ improvement (from $65.5\%$ to $66.7\%$) in top-1 ImageNet accuracy.

With $1\%$ labeled data we find that SuNCEt matches the best SimCLR 500-epoch top-5 accuracy while using only $81\%$ of the compute, and matches the best SimCLR top-1 accuracy while using only $83\%$ of compute.[2] While these savings are significant when considering the overall cost of performing 500-epochs of pre-training on 64 V100 GPUs, we note that the improvements are slightly more modest compared to the $10\%$ labeled data setting. This observation supports the hypothesis that improvements in convergence can be related to the availability of labeled data during pre-training. Table 1 shows the top-1 and top-5 model accuracies with $10\%$ labeled data (left sub-table) and with $1\%$ labeled data (right sub-table).

**Cross-entropy.**  Next we experiment with leveraging labeled samples during pre-training using a cross-entropy loss and a parametric linear classifier (as opposed the non-parametric SuNCEt loss). Similarly to the SuNCEt experiments, we use the same hyper-parameters as in Chen et al. (2020a) for pre-training. Figure 3 reports savings with respect to our SimCLR 1000-epoch baseline; the

---

[2]Note that with $1\%$ labeled data, our 500-epoch re-implementation of SimCLR outperforms the original 1000-epoch results of Chen et al. (2020a).

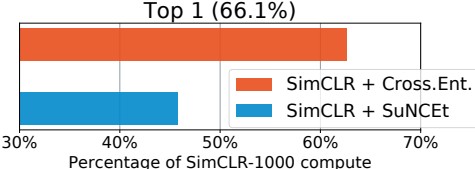 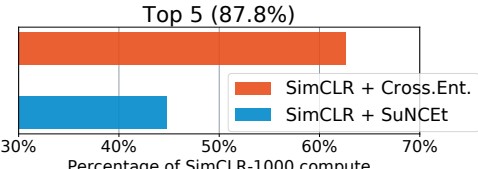

Figure 3: Percentage compute used by various methods to match the best SimCLR (ResNet50, ImageNet) top-1 validation accuracy (top plot) and top-5 validation accuracy (bottom plot) with 1000 epochs of pre-training, followed by fine-tuning with 10% of labels. Both SimCLR + SuNCEt and SimCLR + cross-entropy use the default SimCLR hyper-parameters. The SimCLR+cross-entropy approach matches the best SimCLR validation accuracy while using only 63% of the compute. These savings are lower than those provided by SuNCEt (which only requires ∼44% of pre-training to match the best SimCLR accuracy), but are significant nonetheless.

cross-entropy approach matches the best SimCLR 1000-epoch top-1 and top-5 validation accuracy while using only 63% of the compute. These savings are lower than those provided by SuNCEt (which only requires 44% of pre-training to match the best SimCLR top-5 accuracy and 45% of pre-training to match the best SimCLR top-1 accuracy), but are significant nonetheless.

With 1% labeled data, despite low training loss, SimCLR + cross-entropy does not obtain significantly greater than random validation accuracy with the SimCLR hyper-parameters (even if we only leave the cross-entropy term on for 30 epochs to avoid overfitting). With only 12 samples per class in the 1% data setting, it is quite easy to overfit with a cross-entropy loss, suggesting that more fine-grained tuning may be required. In contrast, recall that we observe 19% compute savings out of the box with SimCLR + SuNCEt in this scenario with the default SimCLR hyper-parameters.

**Transfer.** Our previous results show that leveraging labeled data during pre-training can result in computational savings. Next we investigate the effect of this procedure on downstream transfer tasks. We evaluate transfer learning performance of the 500-epoch pre-trained ImageNet models on Pascal VOC07 (Everingham et al., 2010) (11-mAP), and CIFAR10 and CIFAR100 (Krizhevsky & Hinton, 2009) (top-1), using the fine-tuning procedure described in Chen et al. (2020a) (cf. Appendix B for details). Transfer results are reported in Table 2. Using the SuNCEt loss to leverage available labels during pre-training always improves transfer over pure self-supervised pre-training for the same number of epochs. Moreover, on Pascal VOC07, the SimCLR + SuNCEt combination with only 500 epochs of pre-training significantly outperforms 1000 epochs of SimCLR pre-training.

Table 2: Evaluating transfer learning performance of a ResNet50 pre-trained on ImageNet. Using SuNCEt to leverage available labels during pre-training (not just fine-tuning) always improves transfer relative to self-supervised pre-training with same number of pre-training epochs.

| Method | Epochs | CIFAR10 (Top-1) | CIFAR100 (Top-1) | Pascal VOC07 (11-mAP) |
|---|---|---|---|---|
| SimCLR | 1000 | **97.7** | **85.9** | 84.1 |
| SimCLR | 500 | 97.3 | 84.6 | 83.9 |
| + SuNCEt (ours) 10% *labeled* | 500 | **97.6** | 85.5 | **85.1** |

## 4.2 CIFAR10

**Experimental setup.** Our training setup on CIFAR10 uses a single V100 GPU and 10 CPU cores. We use a learning-rate of 1.0, momentum 0.9, weight decay $10^{-6}$, and temperature 0.5. These hyper-parameters are tuned for SimCLR (Chen et al., 2020a), and we also apply them to the SimCLR + SuNCEt combination. All methods are trained for a default of 500 epochs. Results reported at intermediate epochs correspond to checkpoints from the 500 epochs training runs, and are intended to illustrate training dynamics. We use a batch-size of 256 (512 contrastive samples) for SimCLR. When implementing the SimCLR + SuNCEt combination, we set out to keep the cost per-iteration roughly the same as the baseline, so we use a smaller unsupervised batch-size of 128 (256 contrastive samples) and use a supervised batch-size of 280 (sampling 28 images from each of the 10 classes in the labeled data set). Note that we only sample images from the available set of labeled data to

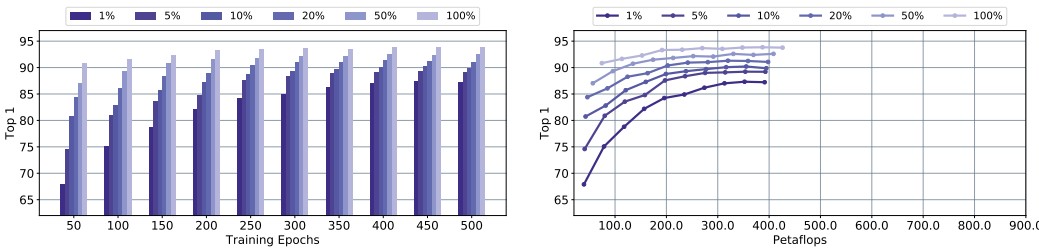

(a) SimCLR test-set convergence with fine-tuning on various percentages of labeled data.

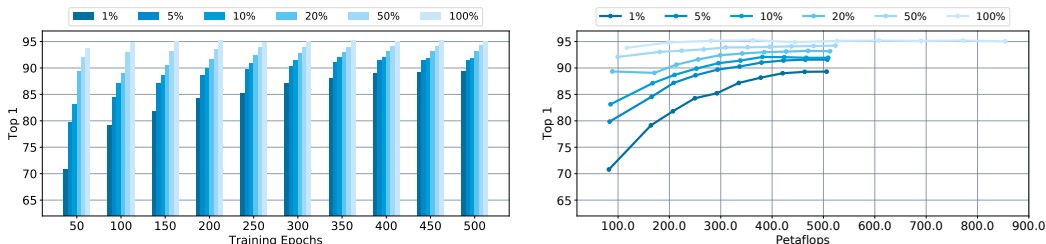

(b) SimCLR + SuNCEt test-set convergence with fine-tuning on various percentages of labeled data.

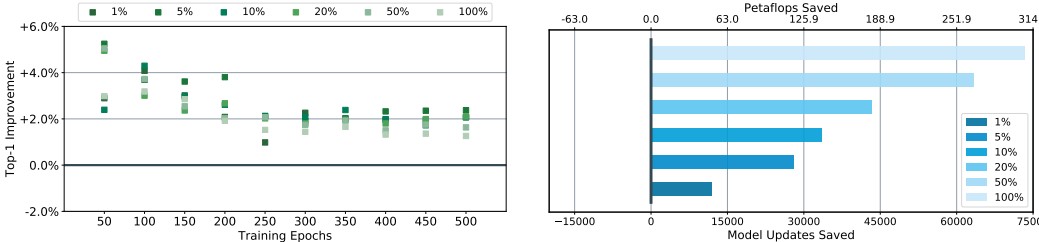

(c) SuNCEt improvement in test-set convergence with fine-tuning on various percentages of labeled data.

(d) Computation saved by SuNCEt in reaching the best SimCLR test accuracy with fine-tuning on various percentages of labeled data.

Figure 4: Training a ResNet50 with an adjusted-stem on CIFAR10 given various percentages of labeled data. Evaluations reported at intermediate epochs correspond to checkpoints from the same run, with a 500 epoch learning-rate cosine-decay schedule. SuNCEt epochs are counted with respect to number of passes through the unsupervised data loader. Both the sample efficiency and computational efficiency of SimCLR improve with the availability of labeled data, even if labeled data is only used for fine-tuning. Using SuNCEt to leverage available labels during pre-training further improves the sample efficiency (sub-figure (c)) and computational efficiency (sub-figure (d)) of SimCLR.

compute the SuNCEt loss in each iteration. We turn off the SuNCEt loss after the first 100 epochs and revert back to completely self-supervised learning for the remaining 400 epochs of pre-training to avoid overfitting to the small fraction of available labeled data; we explore this point in Appendix E.[3]

**Results.** Figure 4a shows the convergence of SimCLR with various amounts of labeled data, both in terms of epochs (left sub-figure) and in terms of computation (right sub-figure). Both the sample efficiency (left sub-figure) and computational efficiency (right sub-figure) of SimCLR improve with the availability of labeled data, even if labeled data is only used for fine-tuning. Figure 4b shows the convergence of the SimCLR + SuNCEt combination with various amounts of labeled data, both in terms of epochs (left sub-figure) and in terms of computation (right sub-figure). Epochs are counted with respect to the number of passes through the unsupervised data-loader. We observe a similar trend in the SimCLR + SuNCEt combination, where both the sample efficiency and computational efficiency improve with the availability of labeled data. Figure 4c shows the improvement in Top-1

---

[3]The only exception to this rule is the set of experiments where 100% of the training data is labeled, in which case we keep SuNCEt on for the entire 500 epochs. We only observed overfitting on CIFAR10, not ImageNet.

test accuracy throughout training (relative to SimCLR) when using SuNCEt during pre-training. Not only does SuNCEt accelerate training from a sample efficiency point of view, but it also leads to better models at the end of training. Figure 4d teases apart the computational advantages by showing the amount of computation saved by the SimCLR + SuNCEt combination in reaching the best SimCLR accuracy. SuNCEt saves computation for any given amount of supervised samples. With only $1\%$ of the training data labeled, SuNCEt can reach the best SimCLR test accuracy while conserving roughly 50 petaflops of computation and over 10000 model updates.[4] In the best case, SuNCEt , with the same exact hyper-parameters as the self-supervised baseline, only requires 22% of SimCLR pre-training to match the best SimCLR test accuracy. It may be possible to push these savings further by optimizing hyper-parameters for SuNCEt .

## 5 RELATED WORK

Table 3: Validation accuracy of a ResNet50 pre-trained on ImageNet with access to 10% of labels. Contrastive methods like SimCLR (Chen et al., 2020a) and SwAV (Caron et al., 2020) can leverage SuNCEt during pre-training to surpass their baseline semi-supervised accuracy in half the number of pre-training epochs. SuNCEt+SwAV is also competitive with other semi-supervised approaches and outperforms FixMatch+RandAugment in terms of top-5 accuracy.

| Method | Epochs | Top-1 | Top-5 |
|---|---|---|---|
| Supervised (Zhai et al., 2019) | 200 | 56.4 | 80.4 |
| NPID++ (Wu et al., 2018b; Misra & van der Maaten, 2020) | 800 | – | 81.5 |
| PIRL (Misra & van der Maaten, 2020) | 800 | – | 83.8 |
| UDA + RandAugment (Xie et al., 2019) | – | 68.8 | 88.5 |
| FixMatch + RandAugment (Sohn et al., 2020) | 300 | **71.5** | 80.1 |
| SimCLRv2 (Chen et al., 2020b) | 1200 | 68.4 | 89.2 |
| SimCLR (Chen et al., 2020a) | 1000 | 65.6 | 87.8 |
| SimCLR+SuNCEt **(ours)** | 500 | 66.7 | 88.2 |
| SwAV (Caron et al., 2020) | 800 | 70.2 | **89.9** |
| SwAV+SuNCEt **(ours)** | 400 | 70.8 | **89.9** |

**Self-supervised learning.** There are a number of other self-supervised learning approaches in the literature, besides the instance-discrimination pretext task in SimCLR (Chen et al., 2020a;b). Some non-contrastive approaches learn feature representations by relative patch prediction (Doersch et al., 2015), by solving jigsaws (Noroozi & Favaro, 2016), by applying and predicting image rotations (Gidaris et al., 2018), by inpainting or colorization (Denton et al., 2016; Pathak et al., 2016; Zhang et al., 2016; 2017), by parametric instance-discrimination (Dosovitskiy et al., 2014), and sometimes by combinations thereof (Doersch & Zisserman, 2017; Kolesnikov et al., 2019). Of the contrastive approaches, Contrastive Predictive Coding (CPC) (Oord et al., 2018; Hénaff et al., 2019) compares representations from neighbouring patches of the same image to produce representations with a local regularity that are discriminative of particular samples. Non-Parametric Instance Discrimination (NPID) (Wu et al., 2018b) aims to learn representations that enable each input image to be uniquely distinguished from the others, and makes use of a memory bank to train with many contrastive samples. The NPID training objective offers a non-parametric adaptation of Exemplar CNN (Dosovitskiy et al., 2014). Misra & van der Maaten (2020) generalizes the NPID method as Pretext-Invariant Representation Learning (PIRL) to contrast images both with and without data augmentations, and combine the method with other instance-wise pretext tasks. He et al. (2019) proposes Momentum Contrast (MoCo) to build even larger memory banks by using an additional slowly progressing key encoder, thus benefiting from more contrastive samples while avoiding computational issues with large-batch training. Grill et al. (2020) also contrasts representations with those of a slowly progressing target encoder, but eliminates negative samples all together. There is also recent work (Li et al., 2020), which makes use of EM (McLachlan, 2004) and clustering algorithms for estimating cluster prototypes (Snell et al., 2017), and the recent SwAV method of Caron et al. (2020), which contrasts image representations with random cluster

---

[4]One model update refers to the process of completing a forward-backward pass, computing the loss, and performing an optimization step.

prototypes. We report results for SwAV+SuNCEt in Table 3, trained with the same exact batch-size and learning-rate as for SuNCEt+SimCLR. The results are consistent with the SimCLR experiments in Section 4; we can match the baseline semi-supervised contrastive accuracy with less than half the pre-training epochs.

**Semi-supervised learning.** Self-supervised learning methods are typically extended to the semi-supervised setting by fine-tuning the model on the available labeled data after completion of self-supervised pre-training. S4L (Zhai et al., 2019) is a recent exception to this general procedure, using a cross-entropy loss during self-supervised pre-training. While Zhai et al. (2019) does not study contrastive approaches, nor the computational efficiency of S4L, it shows that S4L can be combined in a stage-wise approach with other semi-supervised methods such as Virtual Adversarial Training (Miyato et al., 2018), Entropy Regularization (Grandvalet & Bengio, 2006), and Pseudo-Label (Lee, 2013) to improve the final accuracy of their model (see follow-up work (Tian et al., 2020; Hendrycks et al., 2019)). Chen et al. (2020a) reports that the SimCLR approach with self-supervised pre-training and supervised fine-tuning outperforms the strong baseline combination of S4L with other semi-supervised tasks. Other semi-supervised learning methods not based on self-supervised learning include Unsupervised Data Augmentation (UDA) (Xie et al., 2019) and the MixMatch trilogy of work (Berthelot et al., 2019a;b; Sohn et al., 2020). FixMatch (Sohn et al., 2020) makes predictions on weakly augmented images and (when predictions are confident enough) uses those predictions as labels for strongly augmented views of those same images. An additional key feature of FixMatch is the use of learned data augmentations (Berthelot et al., 2019a; Cubuk et al., 2019). Of the non-contrastive methods, FixMatch sets the current state-of-the-art on established semi-supervised learning benchmarks. Note that SwAV+SuNCEt is competitive with the other semi-supervised approaches and outperforms FixMatch+RandAugment in terms of top-5 accuracy (cf. Table 3).

**Supervised contrastive loss functions.** Supervised contrastive losses have a rich history in the distance-metric learning literature. Classically, these methods utilized triplet losses (Chechik et al., 2010; Hoffer & Ailon, 2015; Schroff et al., 2015) or max-margin losses (Weinberger & Saul, 2009; Taigman et al., 2014), and required computationally expensive hard-negative mining (Shrivastava et al., 2016) or adversarially-generated negatives (Duan et al., 2018) in order to obtain informative contrastive samples that reveal information about the structure of the data. One of the first works to overcome expensive hard-negative mining is that of Sohn (2016), which suggests using several negative samples per anchor. Most similar to the SuNCEt loss is that of Wu et al. (2018a), which investigates neighborhood component analysis (NCA) (Goldberger et al., 2005) in the fully supervised setting. However, their method approximates the NCA loss by storing an embedding tensor for every single image in the dataset, adding non-trivial memory overhead. The SuNCEt loss instead relies on noise contrastive estimation and does not have this limitation. Another more recent supervised contrastive loss is that proposed in Khosla et al. (2020) for the fully supervised setting; while their proposed method is more computationally draining than training with a standard cross-entropy loss, it is shown to improve model robustness. ~~As mentioned, the loss can be seen as a form of neighborhood component analysis (Goldberger et al., 2005) with an alternative similarity metric.~~ The SuNCEt loss is different from the loss of Khosla et al. (2020, v1). However, after the initial preprint of our work appeared on OpenReview, Khosla et al. (2020, v2, Section 15-Change Log) was updated with an additional contrastive loss of a similar format to SuNCEt. We provide a brief comparison of the loss in Khosla et al. (2020, v1) and SuNCEt using the full set of labeled data in Appendix H. In short, when using the losses in conjunction with SimCLR and a small supervised batch-size, both methods perform similarly. However, when used independently with larger batches and more positive samples per anchor, their performance differs.

## 6 CONCLUSION

This work demonstrates that a small amount of supervised information leveraged during contrastive pre-training (not just fine-tuning) can accelerate convergence. We posit that new methods and theory rethinking the role of supervision — to not only improve model accuracy, but also learning efficiency — is an exciting direction towards addressing the computational limitations of existing methods while utilizing limited semantic annotations.

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

# Appendix

## A PSEUDO-CODE

Listing 1: Pseudo-code for main training script computing SimCLR+SuNCEt when a small fraction of labeled data is available during pre-training.

```python
# -- init image sampler for instance-discrimination
unsupervised_data_loader = ...

# -- init (labeled) image sampler for SuNCEt
supervised_data_loader = ...

for epoch in range(num_epochs):

    for itr, imgs in enumerate(unsupervised_data_loader):

        # -- compute instance-discrimination loss
        z = mlp(encoder(imgs))
        ssl_loss = simclr(z)

        # -- compute supervised-contrastive loss on labeled data
        imgs, labels = next(supervised_data_loader)
        z = mlp(encoder(imgs))
        supervised_loss = suncet(z, labels)

        # -- compute aggregate loss and update encoder & mlp
        loss = supervised_loss + ssl_loss
        loss.backward()
        optimizer.step()
        lr_scheduler.step()
```

Listing 2: Pseudo-code for computing SuNCEt on a given tensor of image embeddings.

```python
def suncet(z, labels):

    # -- normalize embeddings: [n x d]
    z = z.div(z.norm(dim=1).unsqueeze(1))

    # -- compute similarity between embeddings: [n x n]
    exp_cs = torch.exp(torch.mm(z, z.t()) / temperature).fill_diag(0)

    # -- compute loss for each sampled class and accumulate
    loss = 0.
    num_classes = 0
    for l in set(labels):

        # -- batch-size of embeddings with class-label 'l'
        bs_cls = (labels == l).sum()
        num_classes += 1

        pos_cls = torch.sum(exp_cs[labels == l][:, labels == l], dim=1)
        den_cls = torch.sum(exp_cs[labels == l], dim=1)
        loss += - torch.sum(torch.log(pos_cls.div(den_cls))) / bs_cls

    loss /= num_classes
    return loss
```

## B  ADDITIONAL DETAILS ABOUT FINE-TUNING

We follow the fine-tuning procedure of Chen et al. (2020a). Upon completion of pre-training, all methods are fine-tuned on the available labeled data using SGD with Nesterov momentum. We do not employ weight-decay during fine-tuning, and only make use of basic data augmentations (random cropping and random horizontal flipping). The weights of the linear classifier used to fine-tune the encoder network are initialized to zero. On CIFAR10, models are fine-tuned for 90 epochs. All results are reported on the standard CIFAR10 test set. We use a batch-size of 256, along with a momentum value of 0.9 and an initial learning-rate of 0.05 coupled with a cosine-annealing learning-rate schedule. On ImageNet, in the $10\%$ labeled data setting, models are fine-tuned for 30 epochs; in the $1\%$ labeled data setting, models are fine-tuned for 60 epochs. We use a batch-size of 4096, along with a momentum value of 0.9 and an initial learning-rate of 0.8 coupled with a cosine-annealing learning-rate schedule. All results are reported on the standard ImageNet validation set using a single center-crop.

## C  ADDITIONAL DETAILS ABOUT TRANSFER

We follow the fine-tuning transfer procedure outlined in Chen et al. (2020a). Specifically, we fine-tune the pre-trained model for 20,000 steps using Nesterov momentum. We use a batch-size of 256 and set the momentum value to 0.9. We perform random resized crops and horizontal flipping, and select the learning rate and weight decay by performing grid search with a grid of 7 logarithmically spaced learning rates between 0.0001 and 0.1 and 7 logarithmically spaced values of weight decay between $10^{-6}$ and $10^{-3}$, as well as no weight decay. We divide the weight decay values by the learning rate.

## D  EFFECT OF SUPERVISED BATCH-SIZE ON IMAGENET

What fraction of our sampled mini-batches should correspond to labeled images for computing the SuNCEt loss? We fix the total numbers of passes through the labeled data and vary the fraction of labeled data sampled per mini-batch. Therefore, runs that sample less labeled data per mini-batch keep the SuNCEt loss on for more updates, whereas runs that sample more labeled data per mini-batch keep the SuNCEt loss on for less updates.

We train a ResNet50 on ImageNet for 500 epochs on 64 V100 GPUs using the Sim-CLR + SuNCEt combination with the default Sim-CLR optimization parameters described in Section 4. In one setting, $10\%$ of the images are labeled, and, in the other, $1\%$ of the images are labeled. The left sub-plots in Figure 5 show how the best top-1 and top-5 validation accuracy vary as we change the fraction of labeled data per

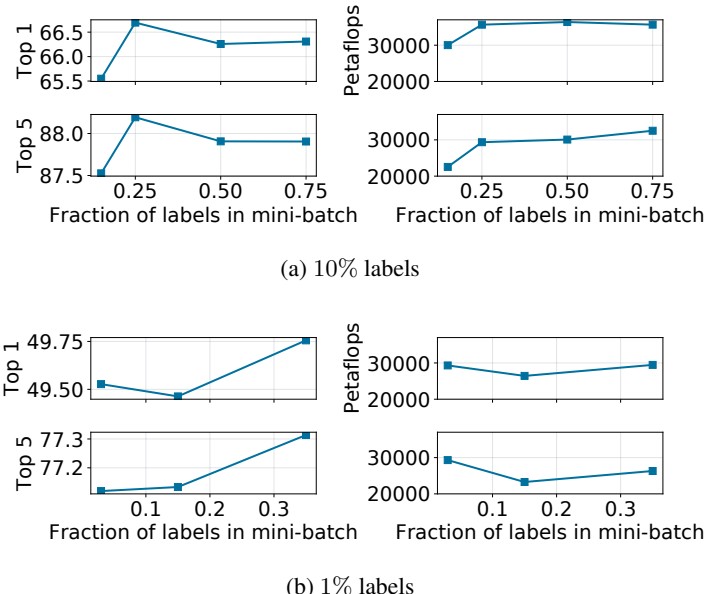

(a) $10\%$ labels

(b) $1\%$ labels

Figure 5: Training a ResNet50 on ImageNet using the SimCLR + SuNCEt combination when given access to $10\%$ of the labels (top sub-plot) and $1\%$ of the labels (bottom sub-plot). We examine the best top-1 and top-5 validation accuracy, and the corresponding computational requirements, as we vary the fraction of labeled samples per mini-batch. If we only use a small fraction of labeled data in each mini-batch, then the best model accuracy drops. However, in general, the best final model accuracies, and the corresponding computational requirements, are not significantly affected by the fraction of labeled data per mini-batch and the corresponding swtich-off epoch.

mini-batch. The right sub-plots show the compute (petaflops) used to obtain the corresponding models in the left subplots vary as we change the fraction of labeled data per mini-batch. If we only use a small fraction of labeled data in each mini-batch, then the best model accuracy drops. However, in general, the best final model accuracies, and the corresponding computational requirements to obtain said models, are not significantly affected by the fraction of labeled data per mini-batch and the corresponding switch-off epoch.

# E   LIMITATIONS ON CIFAR10

The amount of time that we can leave the SuNCEt loss on without degrading performance on CIFAR10 is positively correlated with the amount of labeled data. To shed light on this limitation, we conduct experiments where we switch-off the SuNCEt loss at a certain epoch, and revert to fully self-supervised learning for the remainder training. All models are trained for a total of 500 epochs; epochs are counted with respect to the number of passes through the unsupervised data loader.

The left subplots in Figure 6 report the final model test-accuracy on CIFAR10 as a function of the switch-off epoch, for various percentages of available labeled data. The right subplots in Figure 6 report the amount of petaflops needed to train the corresponding models in the left subplots.

To study the potential accuracy degradation as a function of the switch-off epoch, we first restrict our focus to the left subplots in Figure 6. When 20% or more of the data is labeled (bottom three subplots), the final model accuracy is relatively invariant to the switch-off epoch (lines are roughly horizontal). However, when less labeled data is available, the final model accuracy can degrade if we leave the SuNCEt loss on for too log (top three subplots). The magnitude of the degradation is positively correlated with the amount of available labeled data (lines become progressively more horizontal from top subplot to bottom subplot).

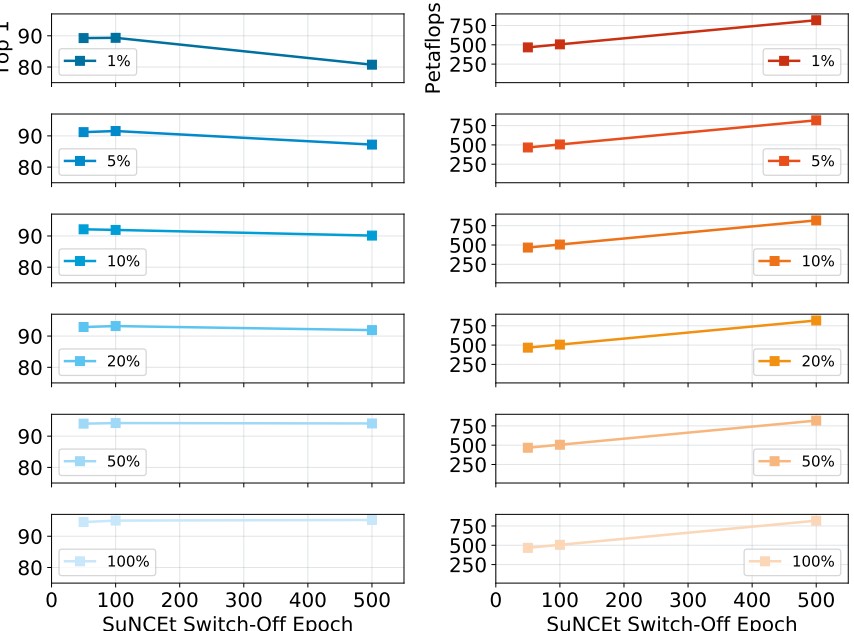

Figure 6: Training a ResNet50 with an adjusted-stem on CIFAR10 given various percentages of labeled data. Left subplots report the final model test-accuracy as a function of the epoch at which the SuNCEt loss is switched off. Right subplots report the amount of petaflops needed to train the corresponding models in the lefts subplots. The number of epochs that one can utilize the SuNCEt loss for without degrading performance is positively correlated with the amount of available labeled data. To balance improvements in model accuracy with computational costs, it may also be beneficial to switch off the SuNCEt loss early on in training, even if leaving it on does not degrade performance.

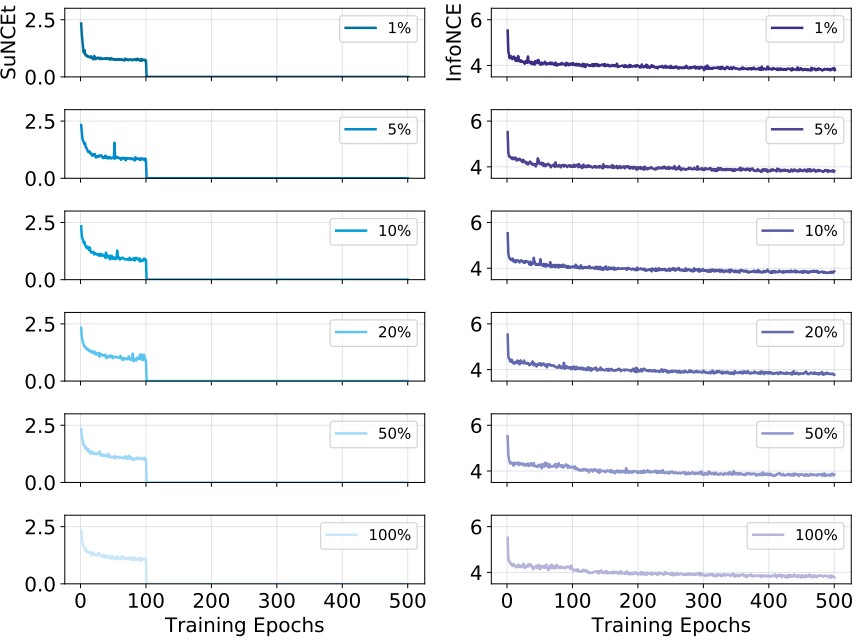

Figure 7: Training a ResNet50 with an adjusted-stem on CIFAR10 given various percentages of labeled data. The SuNCEt loss is only used for the first 100 epochs of training, and then switched off for the reaminder of training. Left subplots report the supervised SuNCEt loss during training. Right subplots report the self-supervised InfoNCE loss during training. In practice, we turn off the SuNCEt loss when it has plateaued, and revert to fully self-supervised training thereafter. In these experiments, the SuNCEt loss has roughly plateaued at around 100 epochs of training.

From a computational perspective, it may also beneficial to turn off the SuNCEt loss at some point, even if leaving it on does not degrade performance. We hypothesize that once we have squeezed out all the information that we can from the labeled data, it is best to revert all computational resources to optimizing the (more slowly-convergent) self-supervised instance-discrimination task. We see that leaving the SuNCEt loss on for more epochs does not provide any significant improvement in model accuracy (left subplots in Figure 6), but the corresponding computational requirements still increase (right subplots in Figure 6).

Switching-off the SuNCEt loss when it has roughly plateaued provides a good strategy for balancing gains in model accuracy with computational costs. We switch off the SuNCEt loss at epoch 100 in all of our CIFAR10 experiments in the main paper (except the experiment with 100% labeled data, where SuNCEt is left on for all 500 epochs of training). Figure 7 depicts the supervised SuNCEt loss during training for various percentages of available labeled data (left subplots), and the self-supervised InfoNCE loss during training for various percentages of available labeled data (right subplots). The SuNCEt loss has roughly plateaued after 100 training epochs (left subplots). Figure 7 also suggests that the rate at which the SuNCEt loss plateaus is negatively correlated with the available amount of labeled data. This observation supports the intuition that one should turn off the SuNCEt loss earlier in training if less labeled data is available (cf. Figure 6). In general, the strategy we adopt is simply to keep the number of passes through the labeled data fixed; meaning that less data will require less updates.

## F    ADDITIONAL EXPERIMENTS FOR SIMCLR + CROSS-ENTROPY (CIFAR10)

**Experimental setup.**    Our training setup for SimCLR + cross-entropy on CIFAR10 is identical to that used in Section 4 for SimCLR + SuNCEt . Specifically, we use a single V100 GPU and 10 CPU cores. We use a learning-rate of $1.0$, momentum $0.9$, weight decay $10^{-6}$, and temperature $0.5$. These hyper-parameters are tuned for SimCLR (Chen et al., 2020a), and we also apply them to the SimCLR + cross-entropy combination.

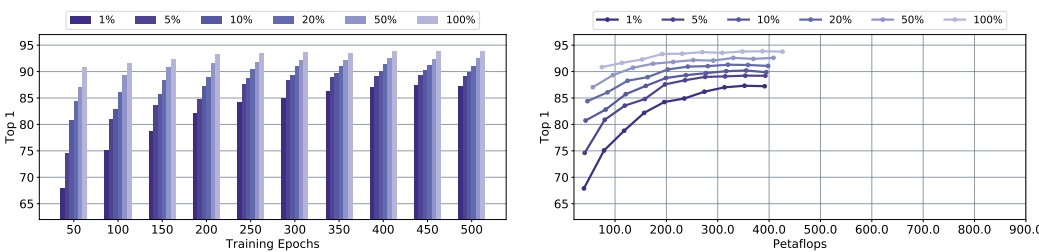

(a) SimCLR test-set convergence with fine-tuning on various percentages of labeled data.

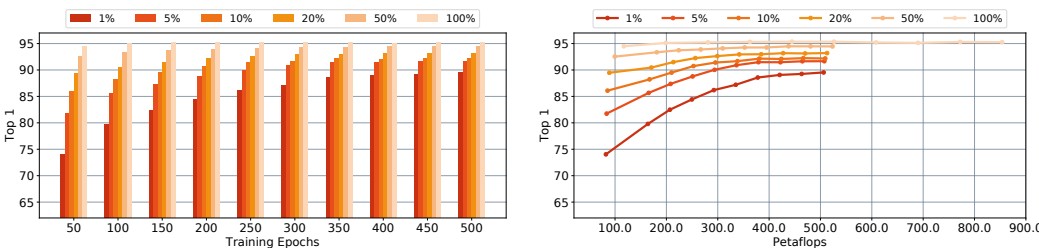

(b) SimCLR + cross-entropy test-set convergence with fine-tuning on various percentages of labeled data.

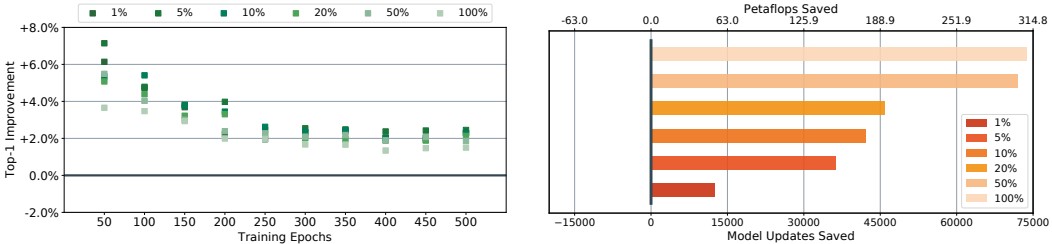

(c) SimCLR + cross-entropy improvement in test-set convergence (relative to plain SimCLR) with fine-tuning on various percentages of labeled data.

(d) Computation saved by SimCLR + cross-entropy in reaching the best SimCLR test accuracy with fine-tuning on various percentages of labeled data.

Figure 8: Training a ResNet50 with an adjusted-stem on CIFAR10 given various percentages of labeled data. Evaluations reported at intermediate epochs correspond to checkpoints from the same run, with a 500 epoch learning-rate cosine-decay schedule. Cross-entropy epochs are counted with respect to number of passes through the unsupervised data loader.

We use a batch-size of 256 (512 contrastive samples) for SimCLR. When implementing the SimCLR + cross-entropy combination, we set out to keep the cost per-iteration roughly the same as the baseline, so we use a smaller unsupervised batch-size of 128 (256 contrastive samples) and use a supervised batch-size of 280 (sampling 28 images from each of the 10 classes in the labeled data set). Note that we only sample images from the available set of labeled data to compute the cross-entropy loss in each iteration.

We turn off the cross-entropy loss after the first 100 epochs and revert back to completely self-supervised learning for the remaining 400 epochs of pre-training to avoid overfitting to the small fraction of available labeled data.[5]

**Results.** Figure 8a shows the convergence of SimCLR with various amounts of labeled data, both in terms of epochs (left sub-figure) and in terms of computation (right sub-figure). Both the sample efficiency and computational efficiency of SimCLR improve with the availability of labeled data, even if labeled data is only used for fine-tuning.

---

[5]The only exception to this rule is the set of experiments where $100\%$ of the training data is labeled, in which case we keep SuNCEt on for the entire 500 epochs.

Figure 8b shows the convergence of the SimCLR + cross-entropy combination with various amounts of labeled data, both in terms of epochs (left sub-figure) and in terms of computation (right sub-figure). Epochs are counted with respect to the number of passes through the unsupervised data-loader. We observe a similar trend in the SimCLR + cross-entropy combination, where both the sample efficiency and computational efficiency improve with the availability of labeled data.

Figure 8c shows the improvement in Top 1 test accuracy throughout training (relative to SimCLR) when using cross-entropy. Similarly to SuNCEt , we see that cross-entropy accelerates training from a sample efficiency point of view, and also leads to better models at the end of training.

Figure 8d shows the amount of computation saved by the SimCLR + cross-entropy combination in reaching the best SimCLR accuracy. There are two x-axes in this figure. The top shows the petaflops saved and the bottom shows the number of model updates saved to reach the best SimCLR test accuracy. Similarly to SuNCEt , cross-entropy saves computation for any given amount of supervised samples. These results provide further evidence for our hypothesis, namely, that leveraging labeled data during self-supervised pre-training can accelerate convergence.

## G    NON-PARAMETRIC INFERENCE

In Section 3 we showed that, from a theoretical perspective, the SuNCEt  loss optimizes a non-parametric classifier based on a type of stochastic nearest neighbours. Here we empirically evaluate this connection on ImageNet by classifying validation images using the inference procedure described in Section 3, and comparing to a K-Nearest Neighbours (KNN) classifier with the same similarity metric.

We consider the $10\%$ labeled data setting and use the 400-epoch pre-trained+fine-tuned Sim-CLR+SuNCEt  models to compute image embeddings. Specifically, we classify each point in the validation set by computing the SuNCEt  class probabilities in equation 3 with respect to the small set of available labeled training images, and choosing the class with the highest probability. We refer to this non-parameteric inference procedure as SuNCEt -NPI. We employ basic data augmentations (random cropping and random horizontal flipping) to the labeled training images before computing their corresponding embeddings, and apply a single center-crop to the validation images. When performing inference using SuNCEt -NPI, we find it best to use the temperature parameter used during training, $\tau = 0.1$ in this case, and, surprisingly, to also use the image embeddings obtained before the MLP projection head. We also find it best to use the image embeddings obtained before the MLP projection head when using the KNN classifier. We experiment with various values of K for the KNN classifier, and find $K = 10$ to work best (surprisingly, better than larger values of K).

Table 4 shows the validation accuracy of these non-parametric classifiers. We consider **(i)** K-Nearest Neighbours (K=10); **(ii)** SuNCEt -NPI (Single-View), where we compute the SuNCEt  class probabilities in equation 3 and use one embedding for each available labeled training image; **(iii)**

Table 4: **Non-parametric inference.** Validation accuracy of a ResNet50 pre-trained on ImageNet for 400-epochs with access to $10\%$ of labels using SimCLR+SuNCEt  with the default SimCLR hyper-parameters. Inference is conducted non-parametrically by computing the similarity of validation images to the $10\%$ labeled train images. We consider **(i)** K-Nearest Neighbours (K=10); **(ii)** SuNCEt -NPI (Single-View), where we compute the SuNCEt  class probabilities in equation 3 and use one embedding for each labeled training image; **(iii)** SuNCEt -NPI (Multi-View), where we compute the SuNCEt  class probabilities in equation 3 and use multiple embedding for each labeled training image. Validation accuracies obtained using SuNCEt -NPI are significantly greater those obtained using K-Nearest Neighbours, suggesting that using SuNCEt  during pre-training optimizes the non-parametric stochastic nearest classifier described in Section 3. Moreover, using multiple views for inference has no significant effect on classification accuracy (likely due to the invariance induced by self-supervised instance-discrimination).

| 10% Labeled | Accuracy (%) | |
| --- | --- | --- |
| **Classification Method** | **Top 1** | **Top 5** |
| KNN (K=10) | 52.3 | – |
| SuNCEt -NPI (Single-View) | **61.7** | **85.8** |
| SuNCEt -NPI (Multi-View) | 61.6 | 85.7 |

SuNCEt -NPI (Multi-View), where we compute the SuNCEt -NPI class probabilities in equation 3 and use multiple embedding for each available labeled training image. The validation accuracies obtained by using SuNCEt -NPI are significantly greater than those obtained using K-Nearest Neighbours; suggesting that using SuNCEt during pre-training optimizes for the non-parametric stochastic nearest classifier described in Section 3.

As a final observation, we find that using multiple views of training images for inference has no significant effect on the classification accuracy; this is likely due to the invariance induced by self-supervised instance-discrimination. It should also be noted that the accuracies in Table 4 are obtained by comparing the validation images to only the $10\%$ of labeled images used during pre-training. It is almost certainly possible to increase the accuracies for all methods in this table by conducting inference with respect to the entire training set. ~~Moreoever~~Moreover, it may be possible to further increase the SuNCEt -NPI accuracies by fine-tuning the pre-trained models using the SuNCEt loss.

## H CONTRASTIVE LOSSES

Here we briefly compare our SuNCEt loss to the loss $\mathcal{L}_{out}^{sup}$ (Khosla et al., 2020), which was proposed for the fully supervised setting. The loss $\mathcal{L}_{out}^{sup}$ with respect to an anchor $z$ with class label $y$ is given by

$$-\frac{1}{|\mathcal{Z}_y(\theta)|} \sum_{z_j \in \mathcal{Z}_y(\theta)} \log \frac{\exp(\text{sim}(z, z_j)/\tau)}{\sum_{z_k \in \mathcal{Z}_{\mathcal{S}}(\theta)\setminus\{z\}} \exp(\text{sim}(z, z_k)/\tau)},$$

whereas the SuNCEt loss is given by

$$-\log \frac{\sum_{z_j \in \mathcal{Z}_y(\theta)} \exp(\text{sim}(z, z_j)/\tau)}{\sum_{z_k \in \mathcal{Z}_{\mathcal{S}}(\theta)\setminus\{z\}} \exp(\text{sim}(z, z_k)/\tau)}.$$

When using the losses in conjunction with SimCLR and a small supervised batch-size, both methods perform similarly. However, when used independently with many positive samples per anchor, their performance differs. We train a ResNet50 for 100 epochs on ImageNet using the full set of labeled data, followed by 15 epochs of fine-tuning the entire network weights with a cross-entropy loss. We use the default SimCLR data-augmentations and hyper-parameters (learning rate $= 4.8$).

Table 5: Validation accuracy of a ResNet50 pre-trained on ImageNet with access to 100% of labels, using the default SimCLR data-augmentations and hyper-parameters (learning rate $= 4.8$). (Left table): training with SimCLR, using an unsupervised batch-size of 4,096 samples and a supervised batch-size of 1,280 samples. (Right table): training using only the supervised losses and a batch-size of 16k (125 classes, 128 instances per class). When using the losses in conjunction with SimCLR and a small supervised batch-size, both methods perform similarly. However, when using the losses independently with many positive samples per anchor, the SuNCEt loss top-1 validation accuracy is more than +10.0% higher.

| Pre-train loss | Top-1 | Top-5 | Pre-train loss | Top-1 | Top-5 |
|---|---|---|---|---|---|
| rand. init | 51.8 | 76.1 | rand. init | 51.8 | 76.1 |
| SimCLR $+ \mathcal{L}_{out}^{sup}$ | **75.4** | **92.9** | $\mathcal{L}_{out}^{sup}$ | 64.4 | 86.5 |
| SimCLR+SuNCEt **(ours)** | **75.4** | **93.0** | SuNCEt **(ours)** | **75.6** | **92.8** |

In the left sub-table in Table 5 we trained both methods jointly with SimCLR, using an unsupervised batch-size of 4,096 samples, and a supervised batch-size of 1,280. Both SuNCEt and $\mathcal{L}_{out}^{sup}$ perform similarly. In the right sub-table in Table 5, we train the methods independently with a batch size of 16K; 125 classes and 128 instances per class. This is similar to the experimental setup in Section 4, where we each mini-batch contains 128 samples per class (2 sampled by each GPU). We had difficulty getting the Khosla et al. (2020) loss to converge with this many positive samples, even when we made the learning-rate small, so we added a batch-normalization (BN) layer before the final layer of the projection head, and this fixed the issue. We evaluated SuNCEt loss both with and without the BN layer, and it did not affect performance, so we left it in for the purpose of comparison. The SuNCEt accuracy is more than +10% higher.

