# OpenReview forum: "Supervision Accelerates Pre-training in Contrastive Semi-Supervised Learning of Visual Representations"
_ICLR.cc/2021/Conference — Reject_

### Official Review · AnonReviewer3 · 2020-10-14
**Official Blind Review #3**

**Rating:** 4
**Confidence:** 5

**Review:**

Summary: This paper combines the self-supervised contrastive loss with the supervised contrastive loss for semi-supervised learning. By leveraging a small amount of labeled data, this paper shows that the semi-supervised contrastive loss can achieve similar performance as self-supervised contrastive loss (SimCLR) with less than half of the compute.

Strength:
1. This paper is well-motivated. It is a good direction to explore contrastive learning in the semi-supervised learning setting.
2. It is interesting to see that using a small amount of labeled data can reduce computation.
3. The paper is well-written and easy to understand.

Weakness:
1. The proposed semi-supervised contrastive loss seems to be a straight-forward combination of the self-supervised contrastive loss and the supervised contrastive loss [Khosla et al. 2020]. Therefore, the technical novelty is limited.
2. This paper does not compare with any of the existing semi-supervised learning methods (e.g. FixMatch), even though they are discussed in the related work. Self-supervised learning has been known to be computation expensive, but this paper also needs to justify that the proposed method is more efficient than existing semi-supervised learning methods.
3. Figure 2 shows that combining CE with SimCLR already achieves good performance. Therefore, a simple baseline would be combining FixMatch with SimCLR. I would expect it to be comparable or even better than the proposed contrastive learning method, because previous papers (e.g. S4L) have shown that adding a self-supervised objective helps with semi-supervised learning.
4. The improvement over SimCLR does not seem to be significant under the same training epochs. Given that labeled data is used, this improvement is mostly expected.  Furthermore, the supervised contrastive loss is turned off after some epochs, which means it plays a less important role. If SuNCEt is also trained for 1000 epochs, would it converge to a similar result as SimCLR?
5. In Table 2, it would be interesting to also see the performance of fully-supervised learning.

---

> ### Author Response · Authors · 2020-11-17
> **Thank you for your feedback**
>
> Thanks you for your detailed review. We address your questions and concerns in the following.
>
> 1. The SuNCEt loss is different from the loss in Khosla et al., 2020 (v1). Their loss is given by
> $$-\frac{1}{|P|} \sum_{p \in P} \log \frac{\exp{sim(z, z_p)}}{\sum_{a \in A} \exp{sim(z, z_a)}}$$
> whereas the SuNCEt loss is given by
> $$\log\frac{\sum_{p \in P} \exp{sim(z, z_p)}}{\sum_{a \in A} \exp{sim(z, z_a)}}.$$
> Their arXiv paper was updated to (v2) on Oct.29 with an additional supervised loss of a similar format to SuNCEt, but this was after our paper was already submitted. They also explicitly mention this in Section 15 of their paper (Change Log).
>
>    We will elaborate on the differences between the loss in Khosla et al., 2020 (v1), and SuNCEt in the appendix. When used in conjunction with SimCLR and a small supervised batch-size, both methods perform similarly. However, when used with many positive samples per anchor their performances differ. We compare a 100-epoch run of the loss in Khosla et al. and SuNCEt on the full set of labeled data using a ResNet-50, followed by 15 epochs of fine-tuning the entire network weights with a cross-entropy loss. Each mini-batch samples 125 classes and 128 instances per class. We use the default SimCLR data-augmentations and hyper-parameters (learning-rate=4.8). This is a similar setup to the experiments in our paper, where we use 128 samples per class (2 sampled by each GPU). We had difficulty getting the Kholsa et al. 2020 loss to converge with this many positive samples, even when we made the learning-rate small, so we added a batch-normalization (BN) layer before the final layer of the projection head, and this fixed the issue. We evaluated SuNCEt both with and without the BN layer, and it did not affect performance, so we left it in for the purpose of comparison. The SuNCEt loss top-1 validation accuracy is more than +10.0% higher.
>     | Pre-train Loss | Top-1 (%) | Top-5 (%) |
> |---|:---:|:---:|
> | rand. init | 51.8 | 76.1|
> |Kholsa et al., 2020 ($\mathcal{L}^{sup}_{out}$)| 64.4 | 86.5 |
> |SuNCEt  **(ours)** | **75.6** | **92.8** |
>
> 2. We're happy to add a table comparing with other methods as well (cf. below; ResNet-50; ImageNet; 10% labels). However, to clarify, our goal is to demonstrate that, in contrast to recent practices in the literature, utilizing available labels during contrastive pre-training, not just fine-tuning, significantly accelerates convergence. We validated this hypothesis with both the SuNCEt loss and a cross-entropy loss, but it was much more pronounced with the SuNCEt loss.
>     | Method | Epochs | Top-1 (%) | Top-5 (%) |
> |--------------------|--------:|:--------:|:------:|
> | Supervised (Zhai et al., 2019) | 200 | 56.4 | 80.4 |
> | NPID++ (Wu et al., 2018) | 800 | -- | 81.5 |
> | PIRL (Misra et al., 2020) | 800 | -- | 83.8 |
> | UDA  + RandAugment (Xie et al., 2020)  | -- | 68.8 | 88.5 |
> | FixMatch + RandAugment  (Sohn et al., 2020) | 300 | **71.5** | 89.1 |
> | (Chen et al., 2020 [b]) | 1200 | 68.4 | 89.2 |
> | -------- |
> | SimCLR (Chen et al., 2020 [a]) | 1000 | 65.6 | 87.8 |
> | SuNCEt+SimCLR  **(ours)** | 500 | 66.7 | 88.2 |
> | -------- |
> | SwAV (Caron et al., 2020) | 800 | 70.2 | **89.9** |
> | SuNCEt+SwAV  **(ours)** | 400 | 70.8 | **89.9** |
>
> 3. We have added a comparison with FixMatch above, and added a comparison with another contrastive method, SwAV (Caron et al., 2020). SuNCEt+SwAV achieves similar state-of-the-art performance to the SwAV baseline in **half** the number of pre-training epochs. SuNCEt+SwAV is also competitive with other semi-supervised approaches and outperforms FixMatch+RandAugment in terms of top-5 accuracy.
>
> 4. The improvements are significant under the same training epochs; e.g. in the 10% label setting, with 100 epochs of training we observe a +1.6% improvement (from 61.8% to 63.4%) in top-1 ImageNet accuracy (run-to-run variation is on the order of 0.1-0.2%); similarly with 500 epochs of training we observe a +1.3% improvement (from 65.4% to 66.7%) in top-1 ImageNet accuracy. We will add a figure making these improvements more clear. There may also have been a confusion; the baseline SimCLR method uses the same amount of labels as SuNCEt. All of the SimCLR runs reported in the figures and the tables fine-tune the entire network weights on the available labels. The point is to show that using labels during pre-training, not just fine-tuning, accelerates convergence. Our speedup on ImageNet saves 48 hours on the equivalent of 8 p3dn.24xlarge machines, which corresponds to savings of $12,000 on AWS for training a single model.
>
>    We do not observe any issues to leaving the SuNCEt loss on for longer on ImageNet, but it did not significantly improve accuracy either. Therefore it is computationally advantageous to switch it off at some point; we reported this ablation in the appendix.
>
> 5. Added in table above (top row).
>
> Thanks again for your review. Please let us know if we have addressed your concerns or if you have other questions.

---

> > ### Comment · AnonReviewer3 · 2020-11-25
> > **Technical novelty and performance improvement not significant enough.**
> >
> > I appreciate the authors' detailed response. However, after reading it, I still feel that this paper is not strong enough in terms of technical novelty or performance improvement. It is not surprising to me that pre-training can accelerate performance. Therefore, I remain my original score.

---

### Official Review · AnonReviewer2 · 2020-10-25
**This submission discovers that supervised contrastive learning can speed up representation learning**

**Rating:** 4
**Confidence:** 5

**Review:**

Pros:

1. This submission is well written with lots of experiments. The claim is that supervised contrastive learning can speed up representation learning, which is well supported.

Cons:

1. Instead of proposing a new idea, this paper adopts supervised contrastive learning and discovers that supervision can help accelerate the pretraining stage. Despite it maybe practical to speed up the experimental cycle, the technique contribution is rather limited. I may understands this paper wrong, so could authors clarify the propsoed SuNCEt loss with supervised contrastive loss?

2. Since this paper is investigating semi-supervised learning, then some comparisons to existing literature is necessary. For example, comparisons to FixMatch. Right now, all the comparisons are only done on SimCLR, which are basically ablation studies.

3. Performance improvement on CIFAR is marginal. I read the limitation section in Appendix, but not clear how it is related to the final results.

4. I notice that the proposed SuNCEt loss is turned off after some training epochs. Especially when using 1% of the labeled data, this loss is turned off at epoch 30, which is quite early in the learn stage. Does SuNCEt loss hurt the training and why? And in practice, how do you determine the optimal time to turn off SuNCEt loss if you don't have access to all the labeled validation set?

In conclusion, despite the paper has some interesting results, it lacks of experiments to show its real contributions.

---

> ### Author Response · Authors · 2020-11-17
> **Thank you for your feedback**
>
> Thank you for taking the time of reviewing our paper.
>
> Indeed, our results on ImageNet save 48 hours on 8 p3dn.24xlarge machines, equivalent to $12,000 on AWS for training a single model. We are the first to demonstrate that leveraging labels during pretraining leads to this improvement in efficiency.
>
> 1. The SuNCEt loss is also different from the supervised contrastive loss in (v1) Khosla et al., 2020. Their loss is given by
> $$-\frac{1}{|P|} \sum_{p \in P} \log \frac{\exp{sim(z, z_p)}}{\sum_{a \in A} \exp{sim(z, z_a)}}$$
> whereas the SuNCEt loss is given by
> $$\log\frac{\sum_{p \in P} \exp{sim(z, z_p)}}{\sum_{a \in A} \exp{sim(z, z_a)}}.$$
> Their arXiv paper was updated to (v2) on Oct.29 with an *additional* supervised loss of a similar format to SuNCEt, but this was after our paper was already submitted. They also explicitly mention this in Section 15 of their paper (Change Log).
>
>    We will elaborate on the differences between the loss in Khosla et al., 2020 (v1), and SuNCEt in the appendix. When using the losses in conjunction with SimCLR and a small supervised batch-size, both methods perform similarly. However, when used independently with larger batches and more positive samples per anchor, their performances differ. We compare a 100-epoch run of the loss in Khosla et al. and SuNCEt on the **full set of labeled data** using a ResNet-50, followed by 15 epochs of fine-tuning the entire network weights with a cross-entropy loss. Each mini-batch samples 125 classes and 128 instances per class. We use the default SimCLR data-augmentations and hyper-parameters (learning-rate=4.8). This is a similar setup to the experiments in our paper, where we use 128 samples per class (2 sampled by each GPU). We had difficulty getting the Kholsa et al., 2020 loss to converge with this many positive samples, even when we made the learning-rate small, so we added a batch-normalization (BN) layer before the final layer of the projection head, and this fixed the issue. We evaluated SuNCEt both with and without the BN layer, and it did not affect performance, so we left it in for the purpose of comparison. The SuNCEt loss top-1 validation accuracy is more than +10.0% higher.
>
>     | Pre-train Loss | Top-1 (%) | Top-5 (%) |
> |---|:---:|:---:|
> | rand. init | 51.8 | 76.1|
> |Kholsa et al., 2020 ($\mathcal{L}^{sup}_{out}$)| 64.4 | 86.5 |
> |SuNCEt  **(ours)** | **75.6** |**92.8** |
>
> 2. We would be happy to add a table comparing with other semi-supervised learning methods as well (cf. table below; ResNet-50; ImageNet; 10% labels). The paper focuses on comparisons to SimCLR because it is the baseline we aim to improve upon. But we have now added a comparison with FixMatch below, and added a comparison with another contrastive method, SwAV (Caron et al., 2020). SuNCEt+SwAV achieves similar performances to the SwAV baseline in *half* the number of pre-training epochs. SuNCEt+SwAV is also competitive with other semi-supervised approaches and outperforms FixMatch+RandAugment in term of top-5 accuracy.
>
>     | Method | Epochs | Top-1 (%) | Top-5 (%) |
> |--------------------|--------:|:--------:|:------:|
> | Supervised (Zhai et al., 2019) | 200 | 56.4 | 80.4 |
> | NPID++ (Wu et al., 2018) | 800 | -- | 81.5 |
> | PIRL (Misra et al., 2020) | 800 | -- | 83.8 |
> | UDA + RandAugment  (Xie et al., 2020)  | -- | 68.8 | 88.5 |
> | FixMatch + RandAugment  (Sohn et al., 2020) | 300 | **71.5** | 89.1 |
> | (Chen et al., 2020 [b]) | 1400 | 68.4 | 89.2 |
> | -------- |
> | SimCLR (Chen et al., 2020 [a]) | 1000 | 65.6 | 87.8 |
> | SuNCEt+SimCLR  **(ours)** | 500 | 66.7 | 88.2 |
> | -------- |
> | SwAV (Caron et al., 2020) | 800 | 70.2 | **89.9** |
> | SuNCEt+SwAV  **(ours)** | 400 | 70.8 | **89.9** |
>
> 3. On CIFAR10 we observe that using labels during pre-training, not just fine-tuning, can reduce training time by up to 88%. This is included as supporting evidence for the ImageNet results.
>
> 4. We also do not observe any issues to leaving the SuNCEt loss on for the entirety of training on ImageNet, but doing so did not significantly improve accuracy, therefore it is computationally advantageous to switch it off at some point; we reported this ablation in the appendix. Note that it is common in modern semi-supervised algorithms to decrease the relative weight of the supervised terms throughout training according to some schedule (cf. [1,2,3]). In practice, we just fixed ahead of time the number of epochs through the labeled data rather than trying to tune a decay schedule. This switch-off approach is also more computationally efficient than a scheduled decay.
> [1] S. Laine and T. Aila, Temporal ensembling for semisupervised learning. ICLR 2017,
> [2] D. Berthelot et al., Mixmatch: A holistic approach to semi-supervised learning. NeurIPS 2019
> [3] D. Berthelot et al., Remixmatch: Semi-supervised learning with distribution matching and augmentation anchoring. ICLR 2020
>
> Thank you again for your review. Please let us know if we have addressed your concerns.

---

### Official Review · AnonReviewer1 · 2020-10-28
**well motivated but needs more work and comparison**

**Rating:** 4
**Confidence:** 3

**Review:**

This paper proposes a new method applicable to a specific case in unsupervised learning: having access to a small amount of labels during the training phase where label signal is not used.

First, I would like to say this is a well motivated task. Many recent works on self-supervised learning (image classification using instance discrimination signals) eventually need to use all the labels in the linear evaluation phase to obtain the final performance numbers. The authors also point out a similar view in the last paragraph in Section 3.

In the related works section, the work by Khosla et al. 2020 is mentioned. I wonder if it makes sense for the authors to add their numbers in the experiment comparison given that Khosla et al also use the original SimCLR as benchmark? In addition, I think a "soft-nearest neighbor loss" paper could be cited and compared (for example: Zhirong Wu, Alexei A Efros, and Stella Yu. Improving generalization via scalable neighbor- hood component analysis. In European Conference on Computer Vision (ECCV) 2018, 2018.)
I understand that most supervised contrastive learning frameworks assume full label during the entire process, which is different from the setting in the paper. But it would be interesting (even critical) to compare the proposed method vs the other supervised contrastive learning methods when SuNCEt is given the full data.

The main contribution of the paper is the proposed SuNCEt loss, which is modified on top of the regular NCE loss in instance discrimination training. However, this “supervised constrastive learning“ only accelerates the SIMCLR learning process (by ~2x in terms of epochs from Table 1 and Table 2) and it does not significantly improve the accuracy over regular SIMCLR.

Overall, I feel that the proposed loss is not a very novel idea (i.e., supervised contrastive learning) and the experiment results are not significantly better than prior arts.

---

> ### Author Response · Authors · 2020-11-17
> **Thank you for your feedback**
>
> Thank you for your feedback and suggestions.
>
> 1. Thank you for pointing out the Wu-Efros-Yu reference. We will be sure to cite it and describe the connection to that work in our revision. As you aptly pointed out, the setting in that paper is different since we focus on semi-supervised learning, but there are other differences as well. The referenced method approximates the NCA loss by storing an embedding tensor for every single image in the dataset, adding non-trivial memory overhead. Our method instead relies on noise contrastive estimation and does not have this limitation.
>
>    We didn’t previously include the Khosla et al. 2020 numbers because they don’t consider semi-supervised learning. We will elaborate on the differences between the loss in Khosla et al., 2020 (v1), and SuNCEt in the appendix. Their loss is given by
> $$-\frac{1}{|P|} \sum_{p \in P} \log \frac{\exp{sim(z, z_p)}}{\sum_{a \in A} \exp{sim(z, z_a)}}$$
> whereas the SuNCEt loss is given by
> $$\log\frac{\sum_{p \in P} \exp{sim(z, z_p)}}{\sum_{a \in A} \exp{sim(z, z_a)}}.$$
> Their arXiv paper was updated to (v2) on Oct.29 with an additional supervised loss of a similar format to SuNCEt, but this was after our paper was already submitted. They also explicitly mention this in Section 15 of their paper (Change Log).
>
>    We have run additional experiments to compare these two losses. When using the losses in conjunction with SimCLR and a small supervised batch-size, both methods perform similarly. However, when used independently with larger batches and more positive samples per anchor, their performance differs. We compare a 100-epoch run of the loss in Khosla et al. and SuNCEt on the **full set of labeled data** using a ResNet-50, followed by 15 epochs of fine-tuning the entire network weights with a cross-entropy loss. Each mini-batch samples 125 classes and 128 instances per class. We use the default SimCLR data-augmentations and hyper-parameters (learning-rate=4.8). This is a similar setup to the experiments in our paper, where we use 128 samples per class (2 sampled by each GPU). We had difficulty getting the Kholsa et al., 2020 loss to converge with this many positive samples, even when we made the learning-rate small, so we added a batch-normalization (BN) layer before the final layer of the projection head, and this fixed the issue. We evaluated SuNCEt both with and without the BN layer, and it did not affect performance, so we left it in for the purpose of comparison. The SuNCEt loss top-1 validation accuracy is more than +10.0% higher.
>
>     | Pre-train Loss | Top-1 (%) | Top-5 (%) |
> |---|:---:|:---:|
> | rand. init | 51.8 | 76.1|
> |Kholsa et al., 2020 ($\mathcal{L}^{sup}_{out}$)| 64.4 | 86.5 |
> |SuNCEt **(ours)** | **75.6** | **92.8** |
>
> 3. We want to clarify that we are not proposing supervised contrastive learning. Our contribution is demonstrating that, in contrast to recent practices in the literature, utilizing available labels during contrastive pre-training, not just fine-tuning, significantly accelerates convergence and maintains/improves downstream transfer. To the best of our knowledge, we are the first  work to demonstrate that effect.
>
>    Respectfully, we would argue that the 2x acceleration is significant. Our speedup on ImageNet saves 48 hours with the equivalent of 8 p3dn.24xlarge machines, corresponding to a savings of $12,000 on AWS for training a single model. Ours is the first work to demonstrate that this savings is possible by leveraging labeled data during pre-training, and not just during fine-tuning, and this is the main contribution of the paper.
>
> Thank you again for your review. Please let us know if we have addressed your concerns.

---

### Official Review · AnonReviewer4 · 2020-10-28
**Involving the labeled samples in pre-training to speed up the contrastive semi-supervised learning**

**Rating:** 6
**Confidence:** 3

**Review:**

**Summary**:
This paper designs a new loss, called SuNCTt, to speed up the convergence of semi-supervised training. Specifically, the loss involves the computation of similarity between anchor and other images with the same class, and the similarity between anchor and other labeled images. It is claimed to be considered as the form of neighborhood component analysis. Together with the standard contrastive learning loss, it only uses less than half the amount of pre-training and computes to match the accuracy of the previous approaches.

**Pros**:
+ The whole idea makes sense. The comprehensive experiments, also shown in appendix, support claims in the paper that the + introduced loss is helpful for semi-supervised training
+ Overall, the paper is well written and the results part is well structured.


**Concerns**:
- It mentions some semi-supervised work in the related work. Is it possible to compare the results with theirs?
- It would be good to show the results of using cross-entropy pre-trained on ImageNet. In addition to the saving compute, I am also curious about the final performance on ImageNet after training for 1000(500) epochs.
- On page 5, it mentioned that, with 1% labeled data, it is not significantly greater than the random accuracy. However, in appendix F (figure 7b), the result of 1% data(CIFAR-10) looks good.  I wonder why it is, if I understand the contexts correctly.
- I also wonder, for the labeled data, what will happen if we use explored metric learning methods, like triplet loss. For the triplet loss, we can leverage labels to define the positive and negative samples. Both triplet loss and SuNCTt may do the similar thing, but in different forms.

Overall, I prefer the rating as above the threshold at the current stage. Hope the authors could address my concerns or questions in the rebuttal period.

---

> ### Author Response · Authors · 2020-11-17
> **Thank you for your feedback**
>
> Thank your for your feedback.
>
> 1. Certainly, we will add a table comparing with other semi-supervised learning methods as well (cf. table below; ResNet-50; ImageNet; 10% labels).
>
>     | Method | Epochs | Top-1 (%) | Top-5 (%) |
> |--------------------|--------:|:--------:|:------:|
> | Supervised (Zhai et al., 2019) | 200 | 56.4 | 80.4 |
> | NPID++ (Wu et al., 2018, Misra et al., 2020) | 800 | -- | 81.5 |
> | PIRL (Misra et al., 2020) | 800 | -- | 83.8 |
> | UDA + RandAugment  (Xie et al., 2020)  | -- | 68.8 | 88.5 |
> | FixMatch + RandAugment  (Sohn et al., 2020) | 300 | **71.5** | 89.1 |
> | (Chen et al., 2020 [b]) | 1400 | 68.4 | 89.2 |
> | -------- |
> | SimCLR (Chen et al., 2020 [a]) | 1000 | 65.6 | 87.8 |
> | SuNCEt+SimCLR  **(ours)** | 500 | 66.7 | 88.2 |
> | -------- |
> | SwAV (Caron et al., 2020) | 800 | 70.2 | **89.9** |
> | SuNCEt+SwAV  **(ours)** | 400 | 70.8 | **89.9** |
>
>    We have now also evaluated SuNCEt with SwAV (Caron et al., 2020), and now achieve state-of-the-art semi-supervised performance. We use the same exact batch-size and learning-rate as for SuNCEt+SimCLR. However as a clarification, our goal is not to propose a new semi-supervised approach. It is to demonstrate that utilizing available labels during contrastive pre-training maintains/improves downstream transfer and significantly accelerates convergence compared to only using labels during fine-tuning, which is the current practice in the literature. Our speedup on ImageNet saves 48 hours on the equivalent of 8 p3dn.24xlarge machines, corresponding to saving $12,000 on AWS for training a single model.
>
> 2. Thank you for this suggestion! We have added the fully supervised result to the above table (top row).
>
> 3. Yes that’s correct. The cross-entropy loss didn’t work out of the box on ImageNet with 1% of labels using the default hyperparameters, but cross-entropy worked fine on CIFAR10. In general, we found ImageNet to be much more sensitive to hyperparameters. Fine-grained search for hyper-parameters could potentially fix this issue for the cross-entropy loss. However, using the SuNCEt loss, we were able to demonstrate computational savings without having to retune the baseline method’s hyperparameters.
>
> 4. We also find it quite interesting to explore metric learning approaches (e.g., triplet loss); the advantage is that computing the loss on these methods is much more efficient than with an N-pair loss. We anticipate however that one would need to perform hard-negative-mining for these methods to be more effective. This investigation is beyond the scope of this work, and we leave it for future work.
>
> Thank you again for your review. Please let us know if we have addressed your concerns.

---

### Author Response · Authors · 2020-11-17
**Summary of Updates**

Dear reviewers and area chair,

We would like to thank you for your detailed reviews. We are pleased to hear that the paper is well motivated and well written (R1, R3, R4) and that the claim that a few labels can speed up representation learning is interesting (R3) and well supported (R4, R2). Indeed, our speedup on ImageNet saves 48 hours on the equivalent of 8 p3dn.24xlarge machines, which corresponds to savings of $12,000 on AWS for training a single model.

We want to point out the following new experiments that we will add in the paper following your feedback:
- (R1, R2, R3) we performed an explicit comparison with the supervised contrastive loss (Khosla et al. 2020).

    The SuNCEt loss is different from the loss in Khosla et al., 2020 (v1). Their loss is given by
$$-\frac{1}{|P|} \sum_{p \in P} \log \frac{\exp{sim(z, z_p)}}{\sum_{a \in A} \exp{sim(z, z_a)}}$$
whereas the SuNCEt loss is given by
$$\log\frac{\sum_{p \in P} \exp{sim(z, z_p)}}{\sum_{a \in A} \exp{sim(z, z_a)}}.$$
Their arXiv paper was updated to (v2) on Oct.29 with an additional supervised loss of a similar format to SuNCEt, but this was after our paper was already submitted. They also explicitly mention this in Section 15 of their paper (Change Log).

  When used in conjunction with SimCLR and a small supervised batch-size, both methods perform similarly. For example, we train a ResNet-50 for 100 epochs on ImageNet using the *full set of labeled data*, followed by 15 epochs of fine-tuning the entire network weights with a cross-entropy loss. We use a SimCLR batch-size of 4,096 samples, and a supervised batch-size of 1280, and use the default SimCLR data-augmentations and hyper-parameters (learning-rate=4.8).
  | Pre-train Loss | Top-1 (%) | Top-5 (%) |
|---|:---:|:---:|
| rand. init | 51.8 | 76.1|
|SimCLR+Kholsa et al., 2020 ($\mathcal{L}^{sup}_{out}$)| 75.4 | 92.9 |
|SimCLR+SuNCEt  **(ours)** | 75.4 | 93.0 |

  However, when used independently with many positive samples per anchor, their performances differ. In this next set of experiments we also train for 100 epochs. Each mini-batch samples 125 classes and 128 instances per class, for an overall batch size of 16k. This is a similar setup to the experiments reported in our paper, where we use 128 samples per class (2 sampled by each GPU).
  | Pre-train Loss | Top-1 (%) | Top-5 (%) |
|---|:---:|:---:|
| rand. init | 51.8 | 76.1|
|Kholsa et al., 2020 ($\mathcal{L}^{sup}_{out}$)| 64.4 | 86.5 |
|SuNCEt  *(ours)* | **75.6** | **92.8** |

- (R2, R3, R4) To go beyond a comparison with SimCLR, we explored the combination of SuNCEt with the added self-supervised swapped cluster prediction loss, SwAV  (Caron et al., 2020). We focus on the semi-supervised ImageNet setting with 10% labels. Our SuNCEt+SwAV run uses the same exact batch-size and learning-rate as for SuNCEt+SimCLR. We report our results below.

  | Method | Epochs | Top-1 (%) | Top-5 (%) |
|---|:---:|---:|---:|
| Supervised (Zhai et al., 2019) | 200 | 56.4 | 80.4 |
| SwAV (Caron et al., 2020) | 800 | 70.2 | **89.9** |
| UDA  + RandAugment (Xie et al., 2020)  | -- | 68.8 | 88.5 |
| FixMatch + RandAugment  (Sohn et al., 2020) | 300 | **71.5** | 89.1 |
| SuNCEt+SwAV  **(ours)** | 400 | 70.8 | **89.9** |

  SuNCEt+SwAV achieves similar performance to the SwAV baseline in half the number of pre-training epochs. SuNCEt+SwAV is also competitive with other semi-supervised approaches and outperforms FixMatch+RandAugment in term of top-5 accuracy.

We will soon update our paper with new experiments. We hope that this will address your concerns, please let us know if you have further questions.

---

### Author Response · Authors · 2020-11-24
**Rebuttal Revision**

Dear reviewers and area chair,

Thank you once for your detailed reviews. We have uploaded the revised paper following your feedback:

- (R2, R3, R4) we have added a comparison with other semi-supervised learning methods, added the performance of fully-supervised learning, and gone beyond the SimCLR comparison by evaluating the combination of SuNCEt with the added self-supervised swapped cluster prediction loss, SwAV (Caron et al., 2020). The conclusions are consistent with the SimCLR experiments. We now achieve state-of-the-art top-5 accuracy on ImageNet with 10% labels, while using less than half the pre-training of the previous state-of-the-art.

- (R1, R2, R3) we have highlighted the differences with the supervised contrastive loss (Khosla et al. 2020), and illustrated these differences with numerical experiments on ImageNet using the full-set of training data in the appendix.

- (R3) we have added a figure showing that the improvements are significant under the same amount of pre-training epochs, despite the fact that both methods use the same amount of labeled data, supporting the argument that using labels during pre-training (not just fine-tuning) can accelerate convergence.

- (R1) we have added the Wu-Efros-Yu reference.

We hope that this has addressed your concerns, please let us know if you have further questions.

---

### Decision · Program_Chairs · 2021-01-07
**Final Decision**

**Decision:**

Reject

**Comment:**

The paper proposes to speed-up self-supervised learning for semi-supervised learning by combining self-supervised pretraining and supervised fine-tuning into a single objective. The proposed supervised loss builds on Neighbourhood Components Analysis and soft nearest neighbor losses. Most reviewers are concerned about the novelty of the approach and the significance of empirical results. I agree with both concerns. I appreciate the comparison between $\log \sum \exp$ and $\sum \log \exp$, but it seems a simpler cross entropy loss also achieves a similar goal (potentially somewhat slower). I believe adding more experiments comparing different supervised loss functions across different architectures can help improve the paper.